# Crash severity analysis of vulnerable road users using machine learning

Md Mostafizur Rahman Komol[1,2]*, Md Mahmudul Hasan[1,2],
Mohammed Elhenawy[1,2], Shamsunnahar Yasmin[1,2], Mahmoud Masoud[1,2‡],
Andry Rakotonirainy[1,2‡]

**1** Centre for Accident Research and Road Safety-Queensland, Queensland University of Technology, Brisbane, Australia, **2** Institute of Health and Biomedical Innovation (IHBI), Queensland University of Technology (QUT), Brisbane, Australia

☉ These authors contributed equally to this work.
‡ MM and AR also contributed equally to this work.
* mdmostafizurrahman.komol@hdr.qut.edu.au

**Data Availability Statement:** The data is restricted and will be made available on request. Dataset is restricted to be open access. The dataset is managed by the Data Analyst Group of the Department of Transport and Main Road

## Abstract

Road crash fatality is a universal problem of the transportation system. A massive death toll caused annually due to road crash incidents, and among them, vulnerable road users (VRU) are endangered with high crash severity. This paper focuses on employing machine learning-based classification approaches for modelling injury severity of vulnerable road users—pedestrian, bicyclist, and motorcyclist. Specifically, this study aims to analyse critical features associated with different VRU groups—for pedestrian, bicyclist, motorcyclist and all VRU groups together. The critical factor of crash severity outcomes for these VRU groups is estimated in identifying the similarities and differences across different important features associated with different VRU groups. The crash data for the study is sourced from the state of Queensland in Australia for the years 2013 through 2019. The supervised machine learning algorithms considered for the empirical analysis includes the K-Nearest Neighbour (KNN), Support Vector Machine (SVM) and Random Forest (RF). In these models, 17 distinct road crash parameters are considered as input features to train models, which originate from road user characteristics, weather and environment, vehicle and driver condition, period, road characteristics and regions, traffic, and speed jurisdiction. These classification models are separately trained and tested for individual and unified VRU to assess crash severity levels. Afterwards, model performances are compared with each other to justify the best classifier where Random Forest classification models for all VRU modes are found to be comparatively robust in test accuracy: (motorcyclist: 72.30%, bicyclist: 64.45%, pedestrian: 67.23%, unified VRU: 68.57%). Based on the Random Forest model, the road crash features are ranked and compared according to their impact on crash severity classification. Furthermore, a model-based partial dependency of each road crash parameters on the severity levels is plotted and compared for each individual and unified VRU. This clarifies the tendency of road crash parameters to vary with different VRU crash severity. Based on the outcome of the comparative analysis, motorcyclists are found to be more likely exposed to higher crash severity, followed by pedestrians and bicyclists.

(Queensland), and they confirm the restriction of the dataset for public access. For anyone to get the data, they need to email individually to the the Data Analyst Group of the Department of Transport and Main Road (Queensland) email address: dataanalysis@tmr.qld.gov.au.

**Funding:** The author(s) received no specific funding for this work.

**Competing interests:** The authors have declared that no competing interests exist.

## Introduction

Road crash is a major health burden globally. More alarmingly, the fatal crash records are reported to rise remarkably across different nations in recent years. For example, in Queensland, Australia, road fatalities are reported to increase 21.5% in 2020 relative to 2019 [1]. However, before 2020, several developed countries were able to achieve a significant reduction in road crash fatalities through multisectoral responses. But the targeted road safety of vulnerable road users (VRU) is still far-reaching. In road safety research, pedestrians, bicyclist, motorcyclist are generally referred to as VRUs. These road users are not protected by an external shell or other external protective measures as motor vehicle occupants are, and hence these groups are prone to get severely injured if involved in a road traffic crash [2]. In Australia, in 2020, it was reported that fatalities for pedestrians were around 12.3% in comparison to all road user fatalities. Moreover, 832 pedestrians were reported to be fatally injured in Australia between the years 2014 through 2018 [3]. On the other hand, 179 bicyclists were reported to be involved in fatal crashes between 2014 to 2018. This number represents 3% of all fatal crashes in Australia [4, 5]. Also, motorcyclists represent 13% of all road fatalities in Australia for the years 2014 through to 2018 [6]. These number clearly signify that VRU safety is a serious road safety concern in Australia, like many other nations around the world.

To improve road safety and reduce such unfortunate events, it is crucial to identify the relevant factors that contribute to crash severity outcomes of VRUs. The contribution of some of these critical factors is likely to vary across different VRU groups, whereas other critical variables might play a similar role. It might be beneficial to compare different critical factors contributing to crash severity outcomes of different VRU groups in order to identify a unified solution in mitigating such unfortunate events. Moreover, comparing crash feature patterns of severity outcomes across different VRU groups might be useful in identifying guidance for road safety education targeting different VRU road user groups. The comparative analysis of VRU crash severity outcomes may contribute towards a broad understanding of the current safety concerns of all VRUs. At the same time, the statistics of crash feature characteristics with high crash severity outcomes could be analysed for identifying several preventive countermeasures, such as engineering, enforcement or educational countermeasures, to reduce the road crash-related trauma of VRU groups [7].

In existing safety literature, machine learning modelling techniques recently have emerged as a promising modelling tool for VRU crash severity classification and analysis on the relationship of road crash features with respect to the severity levels. Employing several machine learning algorithms to analyse critical factors of crash severity outcomes for all VRUs together and separately for each VRU group will allow us to develop our understanding of the importance of these factors for each group, while comparisons across different machine learning modelling techniques will enable us to identify the best-performed models. Analysing the importance and relation of road crash features with respect to VRU crash severity levels will give more intuition to a comparative study of crash severity among different VRU modes separately and for all VRUs together.

With the advancement of the intelligent transportation system (ITS), several improved collision preventions and safety techniques targeted towards improving safety for VRU groups with intersection signal control and vehicle communications have been developed and deployed [8]. However, these technologies are expensive due to the installation and maintenance complexities, and it will be prohibitively expensive to implement these technologies across the entire region. So, it is essential to identify the higher crash severity locations for VRU in prioritising these locations for implementations of these advanced traffic management technologies. Moreover, for further development of these collision prevention technologies,

researchers need extensive information on which crash parameters and elements are likely to contribute towards high VRU crash severity and require a higher focus on improving safety. The direct involvement implementation of countermeasures without the evidence from data analysis may require several trial-and-error, and thus it could cause superficial management in road safety improvement. In such scenarios, road safety management is less likely to be economical. Moreover, a countermeasure might not be effective if location-specific safety concerns are not considered. So, analysing the road crash severity of VRU groups is of utmost importance to inform the road safety improvement framework targeted towards improving VRUs safety.

As such, the overarching aim of this study is to analyse critical features associated with different VRU groups—for pedestrian, bicyclist, motorcyclist and all VRU groups together. The critical factor of crash severity outcomes for these VRU groups is estimated in identifying the similarities and differences across different important features associated with different VRU groups. Specifically, three machine learning algorithms, including (1) Random Forest (RF), (2) Support Vector Machine (SVM) and (3) K-Nearest Neighbour (KNN), is used in classifying the crash severity features for VRU groups. So far, there is no specified rule to apply a particular machine learning model for a specific application [9]. Therefore, most of the research uses multiple machine learning models to the same dataset and use comparative analysis. For example, the study performed by Thanh Noi and Kappas [10] compared the performance of three supervised machine learning models, i.e., KNN, RFC and SVM, in the classification of remote sensing images. The authors found the SVM classifier to be the best performing model in this application, while considering accuracy as performance measure. This indicate that different machine learning models show promises in different applications; however, which classifier is best for a specific application is still not clear.

Specifically, in the existing road safety and crash severity analysis research, several studies have compared the performance between machine learning models where KNN, SVM, and RF models show their performance advantages at different domains [11–13]. Therefore, all of these models bear the significance to be used in machine learning-based analysis for road crash severity. These models perform better in different applications and it is not feasible to conclude superiority of one without comparing their performance for the specific application.

The performances of these models are compared by employing different performance measures, such as accuracy, sensitivity, specificity, precision, F1 test, under the receiver operating characteristic curve (ROC), the area under the ROC curve (AUC) score to identify the best fit models across different VRU groups. The comparisons of critical crash features among different VRU groups are performed by measuring the partial dependency of road crash features with the severity levels. For the analysis, the crash data is sourced from the state of Queensland, Australia, for the years through 2013 to 2019.

The rest of the paper is organised as follows: the literature review section provides an overview of the previous literature on VRU crash severity analysis. The methodology section describes the data while also presenting the data pre-processing and classification models. The results section provides the outcome of the study, and it scopes the performance of the machine learning models, the feature ranking analysis and the crash severity probability analysis. The paper is concluded following the discussion on the outcomes, which includes the interpretation of the research outcome and limitations.

## Literature review

In road safety research, analysis of crash severity outcomes is a mature field. In identifying the critical factors contributing to crash severity outcomes, in existing literature, the application of

the statistical approaches has remained the workhorse. Researchers are implementing different statistical approaches and econometric models to identify the crash features contributing to higher crash severity outcomes [14–22]. Discrete choice model, such as random parameter model, random parameter ordered probit analysis, random parameter logit model, mixed logic model are most common statistical approaches used in different road crash data analysis [23–33]. It is beyond the scope of this study to present a detailed literature review of these statistical modelling. Please see Slikboer, Muir [34] for a detailed literature review on these studies. More recently, several studies have also adopted machine learning-based techniques to identify the important crash features for crash severity [35–38]. With the advancement of artificial intelligence, machine learning-based modelling has become popular in identifying road crash severity factors.

Machine learning models perform better in handling data outliers, noisy and missing data values [37]. With the black box tactics, machine learning models do not need any presumption mathematical function. These models have complex structures with robust learning ability [39, 40]. Even the complex crash severity structures can be quickly interpreted using machine learning models [38, 41–43]. Different machine learning algorithms were compared with different statistical models for road crash severity prediction by analysing the accurate prediction rate. These studies concluded that the machine learning algorithms provide a superior prediction of crash severity [44]. Moreover, recent research based on California read-end crash severity data, prediction accuracy was compared among multinomial logit (MNL), mixed multinomial logit (MMNL), and machine learning algorithm: support vector machine (SVM). The study found that SVM shows better results [45]. Therefore, machine learning-based technique for road crash severity analysis is emerging as a promising modelling technique.

A machine learning approach was used with different algorithms for defining the influence weight of different features for the fatal crash severity of Lebanese Roads [46]. A study on SHARP 2 naturalistic driving data was done to compare the impact of different crash severity features between logistic regression analysis and SVM analysis, where they have found superior outcome from SVM [47]. In another research, crash severity was predicted by four machine learning algorithms with 15 different road crash features. They only analysed and compared the severity models and did not measure the feature ranking or feature relation to crash severity levels. The study developed an improved clustering algorithm to enhance prediction accuracy and significantly improved prediction accuracy [48]. However, the application of machine learning approaches in analysing crash severity outcomes for VRU is limited, and very few studies have focused on VRU crash severity analysis [35, 36, 49]. These studies were also limited by the number of explanatory variables considered in existing machine learning-based crash severity modelling techniques.

VRU Crash severity was predicted in a study using decision tree and ensemble prediction models for bicyclist and pedestrians, where the study found significant prediction improvement from ensemble techniques [49]. For motorcyclist crash severities in Ghana, an analogous machine learning algorithm was demonstrated and compared with a multinomial logit model. Here, the machine learning algorithms were found more precise and reliable in predicting crash severity than the multinomial logit model and found the best performance using the Random Forest model among all the classifiers. The study also calculated feature importance and gain information base on the random forest model to show feature ranking with respect to crash severity, and the location type, time of crash and settlement type features showed the highest ranking among them all [36]. In another research conducted by the same researchers on identifying motorcyclist crash severity in Ghana, they considered three different machine learning algorithms. The study found the Simple Cart model with the best

accuracy and identified some significant factor responsible for motorcyclist crash severity, including location type, settlement type, time of the crash, collision type and collision partner [35].

From the analysis of previous literature, it can be observed that different machine learning algorithms and models were implemented for crash severity analysis of different aspects. However, there are still a significant research gap. First, no studies were found on comparative analysis of VRU crash severity by machine learning modelling either for individual or unified VRU groups. Second, a limited number of road crash parameters were considered as input features in existing machine learning-based VRU crash severity modelling. Even though some studies analysed feature importance and ranking [36], there is no dependency analysis of road crash parameters to show how they are changing with high and low crash severity levels. Lastly, the crash severity mechanism is a complex phenomenon that occurred due to a multitude of factors. Therefore, it is important to consider more explanatory variables, such as crash type and nature, the gender and age group of the VRU, road and environmental conditions. Therefore, it would be worthwhile to perform a comprehensive analysis and compare different road crash parameters based on machine learning-based modelling for all VRUs and for each VRU mode separately (pedestrian, bicyclist, and motorcyclist).

The current study is contributed to the analysis of VRU crash severity of Queensland, Australia, by considering crash data for the years 2013–2019. Seventeen different road crash parameters are considered as input features of machine learning models (all feature information is shown in Table 1). For VRU crash severity classification, three machine learning models are built using RF, SVM and KNN as classifiers, and the best performing model is determined based on the comparison of models' performance measures. This model is further used in measuring the partial dependency of individual crash features for each VRU crash severity modes, showing how the crash severity (dependent variable) changes as the feature (independent variables) changes. This partial dependency analysis represents the impact of road crash parameters over the trend of VRU crash severity levels. This helps in understanding the road crash features responsible for severe crashes for different VRU groups. The outcome of this study is likely to inform road safety countermeasure in improving VRU safety for each VRU group while also identifying a unified safety solution targeting all VRU groups.

## Methodology

### Queensland crash dataset

The crash data for the study is sourced from the official crash database of Queensland collected and compiled by the Department of Transport and Main Roads (TMR). The crash data for VRU groups were collected through for the years 2013 to 2019. The crash data reported in QLD does not record any injury crashes since 2010. Therefore, the data has injury severity levels information for the crashes resulting in casualty only and are reported as four scale injury severity levels—minor injury, medical treatment, hospitalisation, and fatal injury. During the 2013–2019 time period, 69 fatal and 1273 hospitalised injury crashes were reported for the VRU groups under consideration—pedestrian, bicyclist and motorcyclist. Among them, motorcyclist bears the highest percentage in both fatal and hospitalised severity. A crash is defined as fatal when the crash victim dies within 30 days of hospitalisation. The crash victim being admitted to the hospital is defined as hospitalised injury. If the crash victim is admitted to the hospital but released after few hours with treatment, then the injury severity is defined as medically treated injury. If the crash victim evades the crash with first-aid treatment, then it is defined as a minor injury severity level.

**Table 1. Data summary.**

| Crash Severity Features | Description | Code | Motorcyclist | | Bicyclist | | Pedestrian | | Unified VRU | |
|---|---|---|---|---|---|---|---|---|---|---|
| | levels | | Frequency | Percentage | Frequency | Percentage | Frequency | Percentage | Frequency | Percentage |
| Period | | | | | | | | | | |
| Year | 2013 | 0 | 1556 | 13.91% | 782 | 14.51% | 664 | 14.50% | 3002 | 14.19% |
| | 2014 | 1 | 1655 | 14.79% | 832 | 15.44% | 629 | 13.74% | 3116 | 14.73% |
| | 2015 | 2 | 1664 | 14.87% | 724 | 13.43% | 664 | 14.50% | 3052 | 14.42% |
| | 2016 | 3 | 1594 | 14.24% | 769 | 14.27% | 695 | 15.18% | 3058 | 14.45% |
| | 2017 | 4 | 1550 | 13.85% | 798 | 14.81% | 677 | 14.79% | 3025 | 14.30% |
| | 2018 | 5 | 1585 | 14.16% | 710 | 13.17% | 603 | 13.17% | 2898 | 13.70% |
| | 2019 | 6 | 1586 | 14.17% | 775 | 14.38% | 646 | 14.11% | 3007 | 14.21% |
| Month | January | 0 | 976 | 8.72% | 443 | 8.22% | 384 | 8.39% | 1803 | 8.52% |
| | February | 1 | 1067 | 9.54% | 489 | 9.07% | 425 | 9.28% | 1981 | 9.36% |
| | March | 2 | 850 | 7.60% | 339 | 6.29% | 335 | 7.32% | 1524 | 7.20% |
| | April | 3 | 777 | 6.94% | 481 | 8.92% | 345 | 7.54% | 1603 | 7.58% |
| | May | 4 | 811 | 7.25% | 373 | 6.92% | 274 | 5.99% | 1458 | 6.89% |
| | June | 5 | 988 | 8.83% | 485 | 9.00% | 410 | 8.96% | 1883 | 8.90% |
| | July | 6 | 903 | 8.07% | 440 | 8.16% | 434 | 9.48% | 1777 | 8.40% |
| | August | 7 | 852 | 7.61% | 508 | 9.42% | 382 | 8.34% | 1742 | 8.23% |
| | September | 8 | 1077 | 9.62% | 504 | 9.35% | 460 | 10.05% | 2041 | 9.65% |
| | October | 9 | 923 | 8.25% | 456 | 8.46% | 373 | 8.15% | 1752 | 8.28% |
| | November | 10 | 972 | 8.69% | 454 | 8.42% | 393 | 8.58% | 1819 | 8.60% |
| | December | 11 | 994 | 8.88% | 418 | 7.76% | 363 | 7.93% | 1775 | 8.39% |
| Day of Week | Monday | 0 | 1686 | 15.07% | 742 | 13.77% | 804 | 17.56% | 3232 | 15.28% |
| | Tuesday | 1 | 1373 | 12.27% | 726 | 13.47% | 599 | 13.08% | 2698 | 12.75% |
| | Wednesday | 2 | 1741 | 15.56% | 659 | 12.23% | 539 | 11.77% | 2939 | 13.89% |
| | Thursday | 3 | 1855 | 16.58% | 571 | 10.59% | 473 | 10.33% | 2899 | 13.70% |
| | Friday | 4 | 1546 | 13.82% | 873 | 16.20% | 719 | 15.71% | 3138 | 14.83% |
| | Saturday | 5 | 1453 | 12.98% | 968 | 17.96% | 714 | 15.60% | 3135 | 14.82% |
| | Sunday | 6 | 1536 | 13.73% | 851 | 15.79% | 730 | 15.95% | 3117 | 14.73% |
| Hour (Time of Day) | Early morning (midnight–6:30 a.m.) | 0 | 941 | 8.41% | 871 | 16.16% | 417 | 9.11% | 2229 | 10.54% |
| | A.m. peak (6:30 a.m.–9:00 a.m.) | 1 | 1783 | 15.93% | 1677 | 31.11% | 726 | 15.86% | 4186 | 19.78% |
| | A.m. off-peak (9:00–noon) | 2 | 2236 | 19.98% | 608 | 11.28% | 684 | 14.94% | 3528 | 16.67% |
| | P.m. off-peak (noon-4:00 p.m.) | 3 | 3388 | 30.28% | 1226 | 22.75% | 1348 | 29.45% | 5962 | 28.18% |
| | P.m. peak (4:00 p.m.–6:30 p.m.) | 4 | 1892 | 16.91% | 809 | 15.01% | 875 | 19.11% | 3576 | 16.90% |
| | Evening (6:30 p.m.–midnight) | 5 | 857 | 7.66% | 199 | 3.69% | 528 | 11.53% | 1677 | 7.93% |
| Road and Environment Characteristics | | | | | | | | | | |
| Road & Environment Condition | Lighting Condition | 1 | 9136 | 81.64% | 4603 | 85.40% | 3941 | 86.09% | 17566 | 83.02% |
| | Road Condition | 2 | 648 | 5.79% | 450 | 8.35% | 482 | 10.53% | 1407 | 6.65% |
| | Rain wet Slippery | 3 | 1387 | 12.39% | 325 | 6.03% | 142 | 3.10% | 2141 | 10.12% |
| | Atmospheric Condition | 4 | 7 | 0.06% | 9 | 0.17% | 8 | 0.17% | 23 | 0.11% |
| | None | 0 | 12 | 0.11% | 3 | 0.06% | 5 | 0.11% | 21 | 0.10% |

*(Continued)*

**Table 1.** (Continued)

| Crash Severity Features | Description | | Motorcyclist | | Bicyclist | | Pedestrian | | Unified VRU | |
|---|---|---|---|---|---|---|---|---|---|---|
| | levels | Code | Frequency | Percentage | Frequency | Percentage | Frequency | Percentage | Frequency | Percentage |
| Roadway Feature | Intersection and Roundabout | 1 | 6275 | 56.08% | 2258 | 41.89% | 2948 | 64.39% | 11481 | 54.26% |
| | Other Roadway Features | 0 | 4915 | 43.92% | 3132 | 58.11% | 1630 | 35.61% | 9677 | 45.74% |
| Traffic and Speed Jurisdiction | | | | | | | | | | |
| Posted Speed Limit | 0–50 km/hr | 0 | 2300 | 20.55% | 2130 | 39.52% | 2222 | 48.54% | 6652 | 31.44% |
| | 60 km/hr | 1 | 5633 | 50.34% | 2887 | 53.56% | 1989 | 43.45% | 10509 | 49.67% |
| | 70–80 km/hr | 2 | 618 | 5.52% | 179 | 3.32% | 146 | 3.19% | 943 | 4.46% |
| | 80–100 km/hr | 3 | 1131 | 10.11% | 143 | 2.65% | 118 | 2.58% | 1392 | 6.58% |
| | 100–110 km/hr | 4 | 1508 | 13.48% | 51 | 0.95% | 103 | 2.25% | 1662 | 7.86% |
| Speeding Driving Factor | Crashes due to Speeding | 1 | 10662 | 95.28% | 5389 | 99.98% | 4552 | 99.43% | 20603 | 97.38% |
| | Crashes irrelevant to Speeding | 0 | 528 | 4.72% | 1 | 0.02% | 26 | 0.57% | 555 | 2.62% |
| Road User Characteristics | | | | | | | | | | |
| Age Group | 0 to 16 | 1 | 20 | 0.18% | 25 | 0.46% | 32 | 0.70% | 77 | 0.36% |
| | 17 to 24 | 2 | 159 | 1.42% | 850 | 15.77% | 1046 | 22.85% | 2055 | 9.71% |
| | 25 to 59 | 3 | 1970 | 17.61% | 647 | 12.00% | 772 | 16.86% | 3389 | 16.02% |
| | 60 to 75 | 4 | 7908 | 70.67% | 3268 | 60.63% | 1910 | 41.72% | 13086 | 61.85% |
| | 75 up | 5 | 1034 | 9.24% | 514 | 9.54% | 494 | 10.79% | 2042 | 9.65% |
| | unknown | 0 | 99 | 0.88% | 86 | 1.60% | 324 | 7.08% | 509 | 2.41% |
| Region | | | | | | | | | | |
| Road Region | Central Queensland | 0 | 910 | 8.13% | 295 | 5.47% | 270 | 5.90% | 1475 | 6.97% |
| | Downs South West | 1 | 522 | 4.66% | 177 | 3.28% | 214 | 4.67% | 913 | 4.32% |
| | Metropolitan | 2 | 3713 | 33.18% | 2146 | 39.81% | 1864 | 40.72% | 7723 | 36.50% |
| | North Coast and Wide Bay/ Burnett | 3 | 2585 | 23.10% | 971 | 18.01% | 830 | 18.13% | 4386 | 20.73% |
| | North Queensland | 4 | 1341 | 11.98% | 757 | 14.04% | 562 | 12.28% | 2660 | 12.57% |
| | South Coast | 5 | 2119 | 18.94% | 1044 | 19.37% | 838 | 18.30% | 4001 | 18.91% |
| Area Remoteness | Inner Regional | 0 | 2628 | 23.49% | 696 | 12.91% | 750 | 16.38% | 4074 | 19.26% |
| | Major Cities | 1 | 6575 | 58.76% | 3866 | 71.73% | 3152 | 68.85% | 13593 | 64.25% |
| | Outer Regional and Remote Areas | 2 | 1987 | 17.76% | 828 | 15.36% | 676 | 14.77% | 3491 | 16.50% |
| Roadsection Authority | Locally controlled | 0 | 6603 | 59.01% | 4026 | 74.69% | 3400 | 74.27% | 14029 | 66.31% |
| | Not coded | 1 | 5 | 0.04% | 4 | 0.07% | 2 | 0.04% | 11 | 0.05% |
| | State controlled | 2 | 4582 | 40.95% | 1360 | 25.23% | 1176 | 25.69% | 7118 | 33.64% |
| Traffic Condition and Management | | | | | | | | | | |
| Vehicle Condition | Unrestrained | 1 | 10162 | 90.81% | 11 | 0.20% | 4398 | 96.07% | 19573 | 92.51% |
| | Unlicensed | 2 | 166 | 1.48% | 24 | 0.45% | 35 | 0.76% | 244 | 1.15% |
| | Unregistered | 3 | 366 | 3.27% | 71 | 1.32% | 46 | 1.00% | 556 | 2.63% |
| | Vehicle Defect | 4 | 489 | 4.37% | 0 | 0.00% | 90 | 1.97% | 771 | 3.64% |
| | None | 0 | 7 | 0.06% | 5284 | 98.03% | 9 | 0.20% | 14 | 0.07% |
| Driver Condition | Inattentive | 1 | 1874 | 16.75% | 726 | 13.47% | 3869 | 84.51% | 5122 | 24.21% |
| | Fatigued | 2 | 156 | 1.39% | 4 | 0.07% | 10 | 0.22% | 213 | 1.01% |
| | Controller Condition | 3 | 1062 | 9.49% | 253 | 4.69% | 357 | 7.80% | 1782 | 8.42% |
| | Worn Helmet | 4 | 1753 | 15.67% | 210 | 3.90% | 318 | 6.95% | 2712 | 12.82% |
| | None | 0 | 6345 | 56.70% | 4197 | 77.87% | 24 | 0.52% | 11329 | 53.54% |

(*Continued*)

**Table 1.** (Continued)

| Crash Severity Features | Description | | Motorcyclist | | Bicyclist | | Pedestrian | | Unified VRU | |
|---|---|---|---|---|---|---|---|---|---|---|
| | levels | Code | Frequency | Percentage | Frequency | Percentage | Frequency | Percentage | Frequency | Percentage |
| Traffic Control | Flashing amber lights (FL) | 0 | 1 | 0.01% | 1 | 0.02% | 1 | 0.02% | 3 | 0.01% |
| | Give way sign (GWS) | 1 | 1522 | 13.60% | 1405 | 26.07% | 196 | 4.28% | 3123 | 14.76% |
| | Miscellaneous (MC) | 2 | 0 | 0.00% | 0 | 0.00% | 2 | 0.04% | 2 | 0.01% |
| | No traffic control (NT) | 3 | 8133 | 72.68% | 3132 | 58.11% | 3147 | 68.74% | 14412 | 68.12% |
| | Operating traffic lights (OTL) | 4 | 1139 | 10.18% | 568 | 10.54% | 758 | 16.56% | 2465 | 11.65% |
| | Pedestrian crossing sign (PCS) | 5 | 25 | 0.22% | 81 | 1.50% | 253 | 5.53% | 359 | 1.70% |
| | Pedestrian operated lights (POL) | 6 | 5 | 0.04% | 12 | 0.22% | 95 | 2.08% | 112 | 0.53% |
| | Police (PL) | 7 | 18 | 0.16% | 2 | 0.04% | 19 | 0.42% | 39 | 0.18% |
| | Railway—lights and boom gate (RL&BG) | 8 | 5 | 0.04% | 6 | 0.11% | 2 | 0.04% | 13 | 0.06% |
| | Railway—lights only (RL) | 9 | 4 | 0.04% | 0 | 0.00% | 0 | 0.00% | 5 | 0.02% |
| | Railway crossing sign (RCS) | 10 | 0 | 0.00% | 1 | 0.02% | 1 | 0.02% | 1 | 0.00% |
| | Road/Rail worker (RW) | 11 | 8 | 0.07% | 4 | 0.07% | 40 | 0.87% | 52 | 0.25% |
| | School crossing—flags (SCF) | 12 | 0 | 0.00% | 0 | 0.00% | 1 | 0.02% | 1 | 0.00% |
| | Stop sign (SS) | 13 | 329 | 2.94% | 177 | 3.28% | 55 | 1.20% | 561 | 2.65% |
| | Supervised school crossing (SSC) | 14 | 1 | 0.01% | 1 | 0.02% | 8 | 0.17% | 10 | 0.05% |
| Traffic Law Impairment | | | | | | | | | | |
| Disobey Road Rule | All driver | 1 | 5680 | 50.76% | 2897 | 53.75% | 3439 | 75.12% | 11438 | 54.06% |
| | Traffic Driver | 2 | 225 | 2.01% | 56 | 1.04% | 76 | 1.66% | 394 | 1.86% |
| | No Giveaway | 3 | 2218 | 19.82% | 1844 | 34.21% | 566 | 12.36% | 4569 | 21.59% |
| | Other road rule violation | 4 | 2958 | 26.43% | 592 | 10.98% | 495 | 10.81% | 4609 | 21.78% |
| | None | 0 | 109 | 0.97% | 1 | 0.02% | 2 | 0.04% | 148 | 0.70% |
| Drink Drug Alcohol Related | Alcohol Drug Related | 0 | 9873 | 88.23% | 5206 | 96.59% | 3927 | 85.78% | 18865 | 89.16% |
| | Drink Driving | 1 | 1220 | 10.90% | 184 | 3.41% | 309 | 6.75% | 1947 | 9.20% |
| | Alcohol Impaired Pedestrian | 2 | 97 | 0.87% | 0 | 0.00% | 342 | 7.47% | 346 | 1.64% |
| Classification Target | | | | | | | | | | |
| Crash Severity | High Crash Severity | 1 | 3987 | 35.63% | 2909 | 53.97% | 1876 | 40.98% | 8772 | 41.46% |
| | Low Crash Severity | 0 | 7203 | 64.37% | 2481 | 46.03% | 2702 | 59.02% | 12386 | 58.54% |

## Data pre-processing

Data pre-processing and cleaning is an important preliminary step for the downstream analysis using machine learning algorithms. The raw data had 65 different features having 21,989 data points. The data were filtered in several steps. First, all the crash data related to VRU and their associated crash events were filtered (bicyclists, motorcyclists, and pedestrians). Second, one of the features from the features containing similar information was considered. Last, the manual check was done on the whole dataset, and finally, 17 features were extracted. The final dataset has a record of 21,158 VRU crashes, including 11,190 data records for motorcyclists, 5,390 records for bicyclist and, 4,578 records for pedestrian for the years 2013 through to 2019.

For our research purpose of classifying the distinction between the major crash injuries and minor crash severity condition of different road crash features, the injury severity levels are

divided into two broad categories—low severity (including minor and medically treated injuries) and high severity (including hospitalised and fatal injuries). The four different severity levels are associated with data imbalance since there are few data records for fatal (572) and minor injury (2000) categories, while the data records are higher for the hospitalised (11814) and medically treated (6772) injury severity categories. The imbalance in the datasets may result in the deteriorated performance of the classification models having an imbalanced confusion matrix and thus a higher gap between sensitivity and specificity. Therefore, to tackle the data imbalance issue in injury severity levels, these were converted to a binary class—low severity (positive class) and high severity (negative class). Such aggregation of injury severity categories helps in clarifying confident viewpoints of road crash features to show their distinction in high and low crash severity levels. The final dataset had 21,158 VRU data records, including 12,386 high severity and 8,772 low severity cases.

The filtered VRU data is further divided into three sub-datasets based on the VRU types—pedestrians, bicyclists, and motorcyclists. In the data, all information of feature and classes are given in descriptive view. However, to work on the methodological procedure, the features were transformed using a label encoder system in Python 3.7.7 platform. The levels of each feature are converted to numerical value counting from zero (0) to the maximum number of levels minus one. Some similar features of the data were incorporated into a single feature to ensure improved performance of classification and to demonstrate their behaviour sequentially. The numerical denotation of selected features of pre-processed data for VRU crash severity is showed in Table 1.

## Classification approaches

The classification approach defines the crash severity classification methodology, outputs, and comparison among different classifiers for unified VRU and for different VRU groups separately: motorcyclists, bicyclists and pedestrians. Followed by the pre-processing stage, the dataset is divided into three different mediums for each individual type of VRU, and the ensemble pre-processed data was set for unified VRU. Three machine learning algorithms: RF, SVM and KNN, are used as classifiers to classify the crash severity for unified VRU and individual motorcyclists, bicyclists, pedestrians. Most of the statistical models work to infer the relationship between or among two or more variables, whereas the machine learning models deal with making the best accurate predictions [50]. The predictor outcomes are used to identify the class of the target variables. On the one hand, the statistical models can handle very small datasets, whereas the machine learning models are designed to deal with big data [51]. The dataset contains 21158 data points, having 17 features. Using statistical methods, such as ANOVA F Test or Correlation-based methods, could make the interpretation complex; thus, we have used machine learning models in this study. Given the different machine learning-based predictive models works on their specific principles, it is worth comparing their performances on the same dataset. The three selected models are supervised learning algorithms. SVM is a linear model that identifies the best classification hyperplane to separate data into desired classes [52]. KNN algorithm analyses the similarity of the data to classify different classes [53]. It considers a K value at the nearest neighbour to the data point for classification. The random forest acts as an ensemble method bagging multiple decision trees for classification [54]. The decision tree is an algorithm containing series of trees with the binary decision about the class. As such, random forest outperforms decision tree as an ensemble technique, decision tree model is not separately used for our crash severity analysis.

The whole data was split into train and test data by 3:1 ratio, where 75% of the data was used to train the models after necessary hyperparameter tuning with cross-validation. Holdout

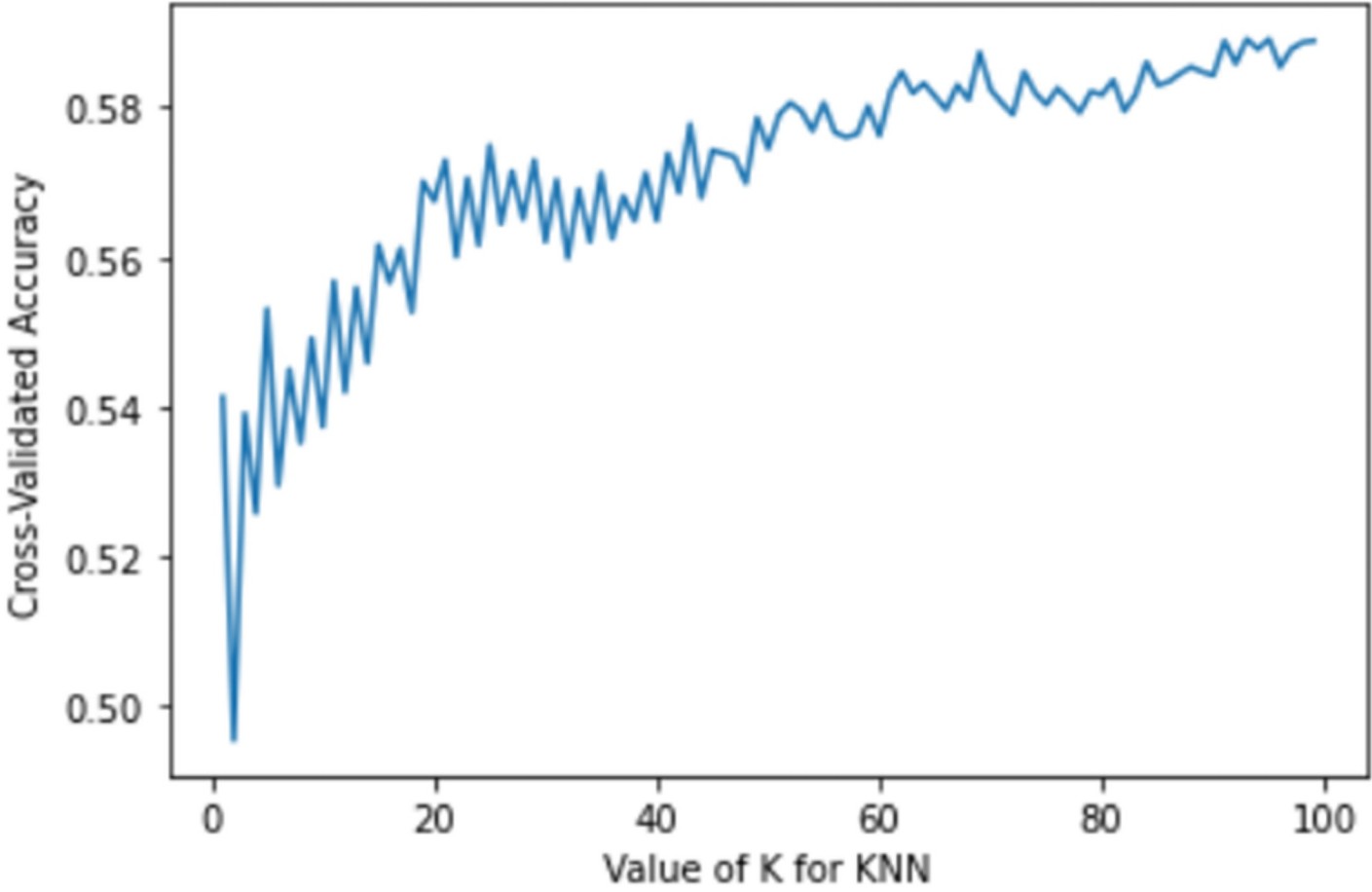

**Fig 1. Hyperparameter tuning in KNN.**

validation was done with the remaining 25% data, which was used for the testing purpose. The hyperparameters of each classifier were set using a consecutive iteration process for precise gradient optimisation, reduction of the loss function and increase the accuracy as well. For the KNN classifiers, a set of the best value of 'k' was used to train the model, and the 'k' value corresponding to the best training accuracy was chosen. For example, the plot for the KNN model of training accuracy for the pedestrian group with different values of 'k' is illustrated in Fig 1.

For the gaussian SVM model, 'rbf kernel' was used, and 'grid-search' for 'C' and 'γ' parameters using the 'grid-search' algorithm and the best values were chosen for the best accuracy after cross-validation. For the random forest classifier models, a grid-search was done on the 'n_estimate' and max depth to tune the number of trees in the forest and the maximum depth of the tree, respectively. Different performance measures are used to evaluate the performance of machine learning models, including sensitivity (true positive rate), specificity (true negative rate), accuracy, area under the receiver operating characteristic curve, precision, and f1 score.

## Analysis results

### Performance of the machine learning models

For model performance measures, accuracy, sensitivity, specificity, precision, F1 test, under the receiver operating characteristic curve (ROC), area under the ROC curve (AUC) score

**Table 2. Performance of classification models for crash severity levels.**

| Motorcyclist | | | |
|---|---|---|---|
| Performance Metrics | Random Forest | Support Vector machine | K-Nearest Neighbour |
| Accuracy | **72.30%** | 68.38% | 65.79% |
| F1 Score | 80.25% | 78.24% | 77.59% |
| Sensitivity (True Positive Rate) | 94.53% | 94.12% | 92.81% |
| Specificity (True Negative Rate) | 29.78% | 17.51% | 13.53% |
| Precision Score | 70% | 67% | 67.45% |
| AUC Score | 0.74 | 0.70 | 0.67 |
| **Bicyclist** | | | |
| Performance Metrics | Random Forest | Support Vector machine | K-Nearest Neighbour |
| Accuracy | **64.45%** | 60.25% | 58.21% |
| F1 Score | 67.15% | 47.69% | 37.88% |
| Sensitivity (True Positive Rate) | 53.53% | 40.95% | 30.22% |
| Specificity (True Negative Rate) | 70.85% | 75.12% | 77.36% |
| Precision Score | 75.87% | 54.23% | 52.33% |
| AUC Score | 0.66 | 0.62 | 0.60 |
| **Pedestrian** | | | |
| Performance Metrics | Random Forest | Support Vector machine | K-Nearest Neighbour |
| Accuracy | **67.23%** | 63.28% | 61.75% |
| F1 Score | 79.35% | 76.49% | 69.2% |
| Sensitivity (True Positive Rate) | 92.66% | 98.12% | 88.56% |
| Specificity (True Negative Rate) | 27.38% | 12.22% | 19.38% |
| Precision Score | 61.67% | 63.5% | 63.7% |
| AUC Score | 0.68 | 0.64 | 0.65 |
| **Unified VRU** | | | |
| Performance Metrics | Random Forest | Support Vector machine | K-Nearest Neighbour |
| Accuracy | **68.57%** | 65.59% | 62.56% |
| F1 Score | 75.35% | 73.67% | 70.72% |
| Sensitivity (True Positive Rate) | 83.56% | 82.23% | 75.98% |
| Specificity (True Negative Rate) | 45.28% | 38.32% | 40.21% |
| Precision Score | 69.37% | 66.23% | 66.31% |
| AUC Score | 0.70 | 0.67 | 0.64 |

were compared comprehensively. The result of different classification model for different VRU types is illustrated sequentially in following Table 2.

From the results and scores of different models for different VRU, it can be observed that the Random Forest classifier shows the best result with model accuracy of 72.30%, 64.45%, 67.23% and 68.57%, respectively, for motorcyclists, bicyclists, pedestrians and unified VRU. For a clear comparison of different machine learning models for different VRU, a bar graph with measurement detail is presented in Fig 2.

Fig 2 represents the performance of the three different classifiers for four different scenarios, i.e. motorcyclists, bicyclists, pedestrians and unified VRU. The output of different classifiers is coded through specific colours. For RF, SVM and KNN, the coloured green, yellow and red bars are used, respectively.

From the performance measure bars of motorcyclists, RF's test accuracy and F1 score are the highest. Nonetheless, SVM generates the nearest sensitivity score (94.12%) compared to 94.53% when using RF. The specificity score for RF (29.79%) is much higher in differences in comparison to SVM specificity (17.51%) for motorcyclists, and RF is the best in precision

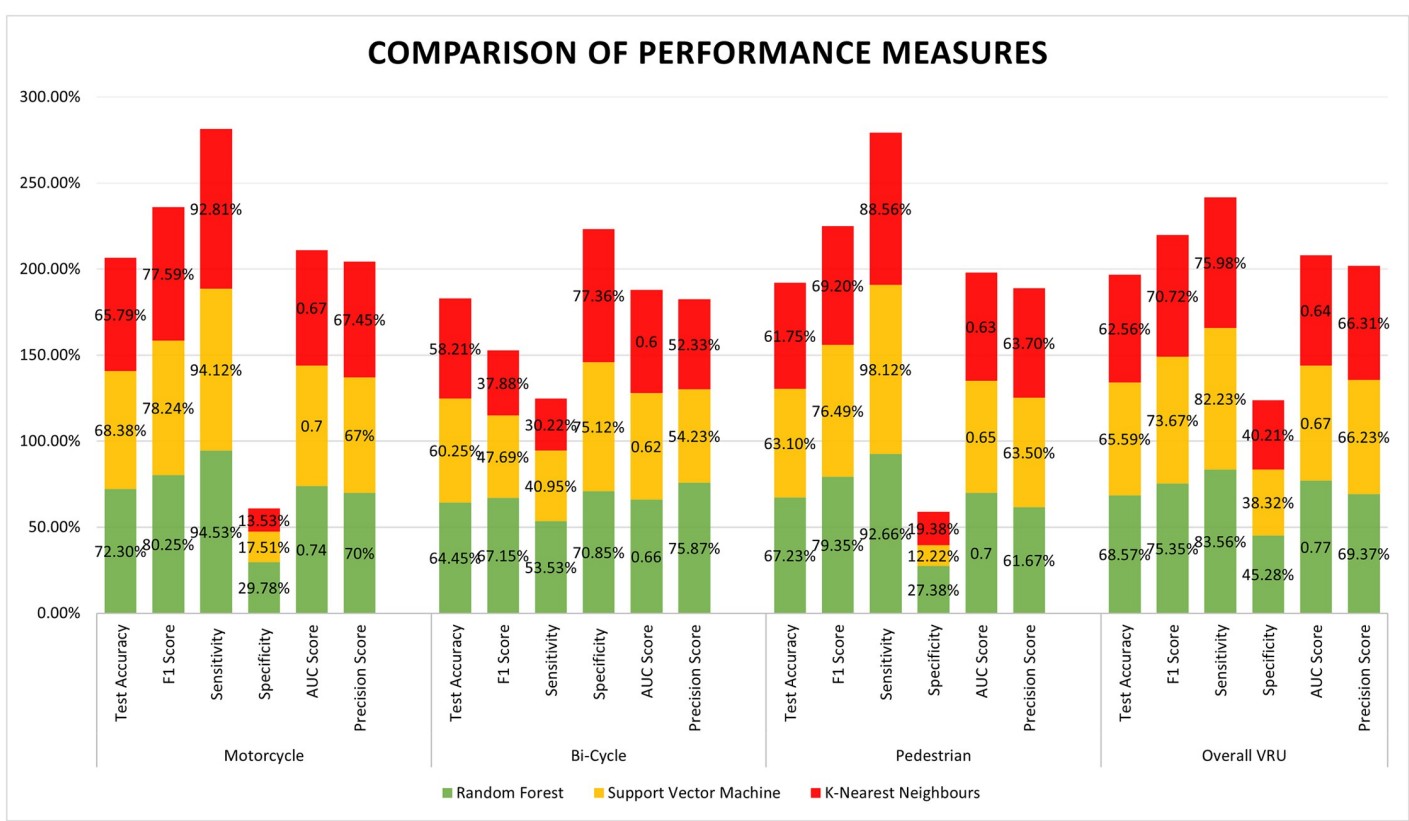

**Fig 2. Comparison performance measures of classifiers for different VRU models.**

(70%) too. For Bicyclists, RF is advanced in test accuracy (64.45%) and other performance parameters than other classifiers. The f1 score (67.15%) and precision score (75.87%) is drastically high for RF than the second largest f1 score (47.69%) and precision (54.23) of SVM. The KNN specificity score (77.36%) shows the highest score, and on the other hand, the RF sensitivity (53.53%) shows the high priority. However, the difference between RF sensitivity with KNN sensitivity and the difference between RF specificity with KNN specificity is almost equivalent.

For Pedestrians, the RF classifier has the best accuracy, f1 score, test accuracy, precision and specificity. The best sensitivity score (98.12%) is achieved from SVM, but the unified score of RF is mostly above the other two classifiers' scores. In the case of unified VRU integrating all motorcyclist, bicyclist and pedestrian, the classification model was generated with RF, SVM and KNN too. Analysing their results, only the sensitivity of SVM (82.23%) is very close to the sensitivity of RF (83.56%). Apart from that, the RF classifier model for unified VRU is found supreme.

Based on comparison and analysis of all classification model for each individual and unified VRU, it is identifiable that the RF classifier is superior to other classifiers in unified performance measure comparison for different VRU types. But still, SVM and KNN classifiers have some advanced scores in a specific field than RF classifier for different VRU groups. So, before pronouncing the RF model as most legitimate and accepted in classifying VRU crash severity, further evaluation step is proceeded by comparing the receiver operating characteristic (ROC) curve of different classifier models for different VRU types. The ROC curve with true positive

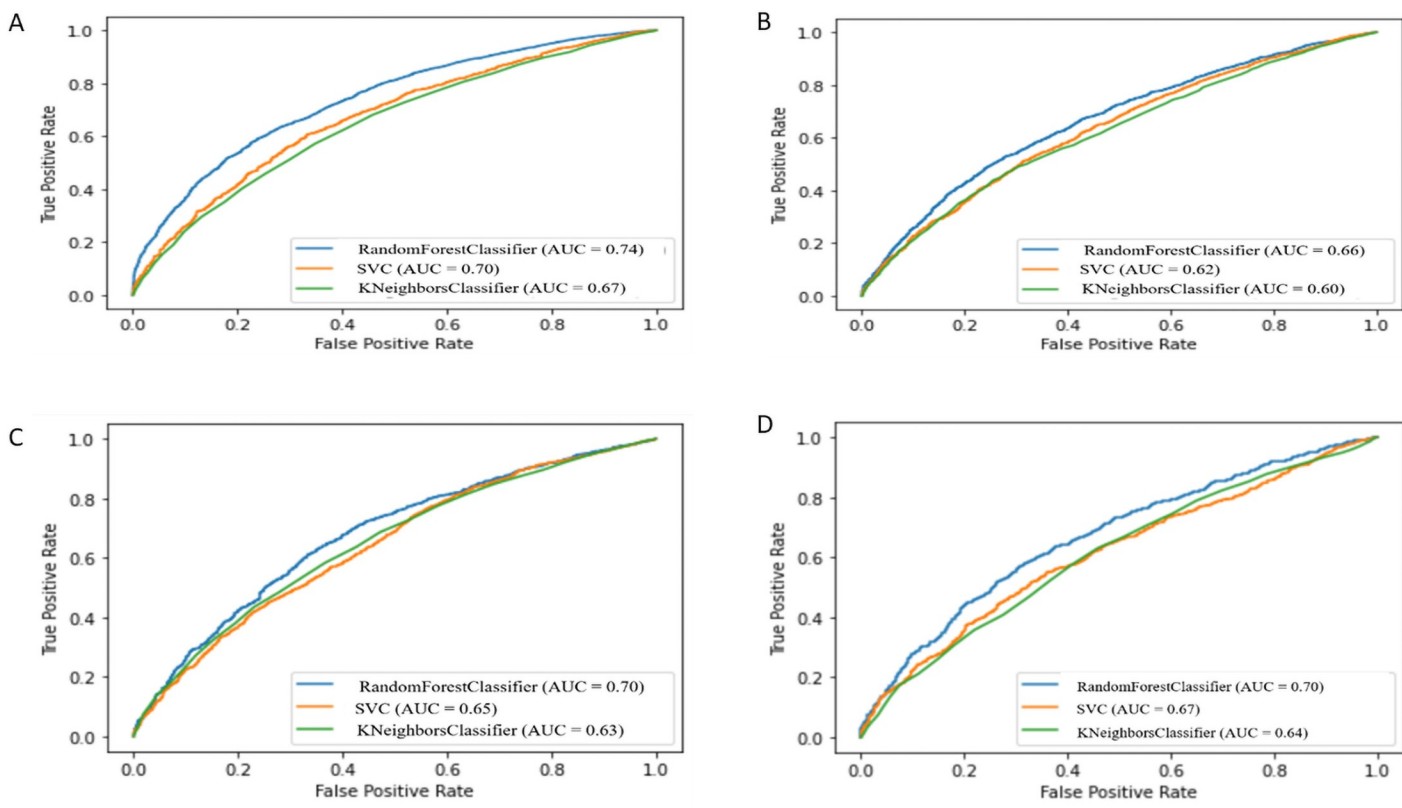

**Fig 3. ROC curves of different classifiers.** (A) Motorcyclists (B) Bicyclists (C) Pedestrians (D) VRUs.

rate and false positive rate for different VRU types by different classifier models are shown in Fig 3.

In Fig 3, the ROC curves of RF, SVM, KNN classifiers are illustrated using blue, orange and green curves, respectively, for each VRU type. From the curves of three individual VRU and unified VRU, it is observed that the RF AUC score for all VRU categories is higher in comparison to other classifiers, and it is around 70% for all VRU except cyclist (66%). As the AUC shows the best compromise between the true positive rate and false positive rate, the higher value of AUC is always preferable while considering a machine learning algorithm. After analysing ROC curves for all VRU groups, it is clearly understandable that the RF classification model outperforms the other three classifiers (KNN, SVM and ANN) in the current study context. So, RF was considered as most authentic and legitimate among other classifiers for modelling the classification of QLD VRU crash severity levels for the years 2013 through 2019. Thus, further analysis, as presented in the following sections, of QLD VRU feature impact for all different VRU types and feature behaviour with the crash severity level was proceeded using the RF classification model.

## Feature ranking analysis

Features ranking refers to the response of each feature to vary with VRU crash severity classification. The feature ranking is done using the random forest feature importance algorithm. As the random forest is a tree-based model, each of the nodes of the decision tree works as a condition for a specific feature, and thus the similar values ended up being listed in the same set. The measure for feature ranking is called 'impurity', and during the training phase of the

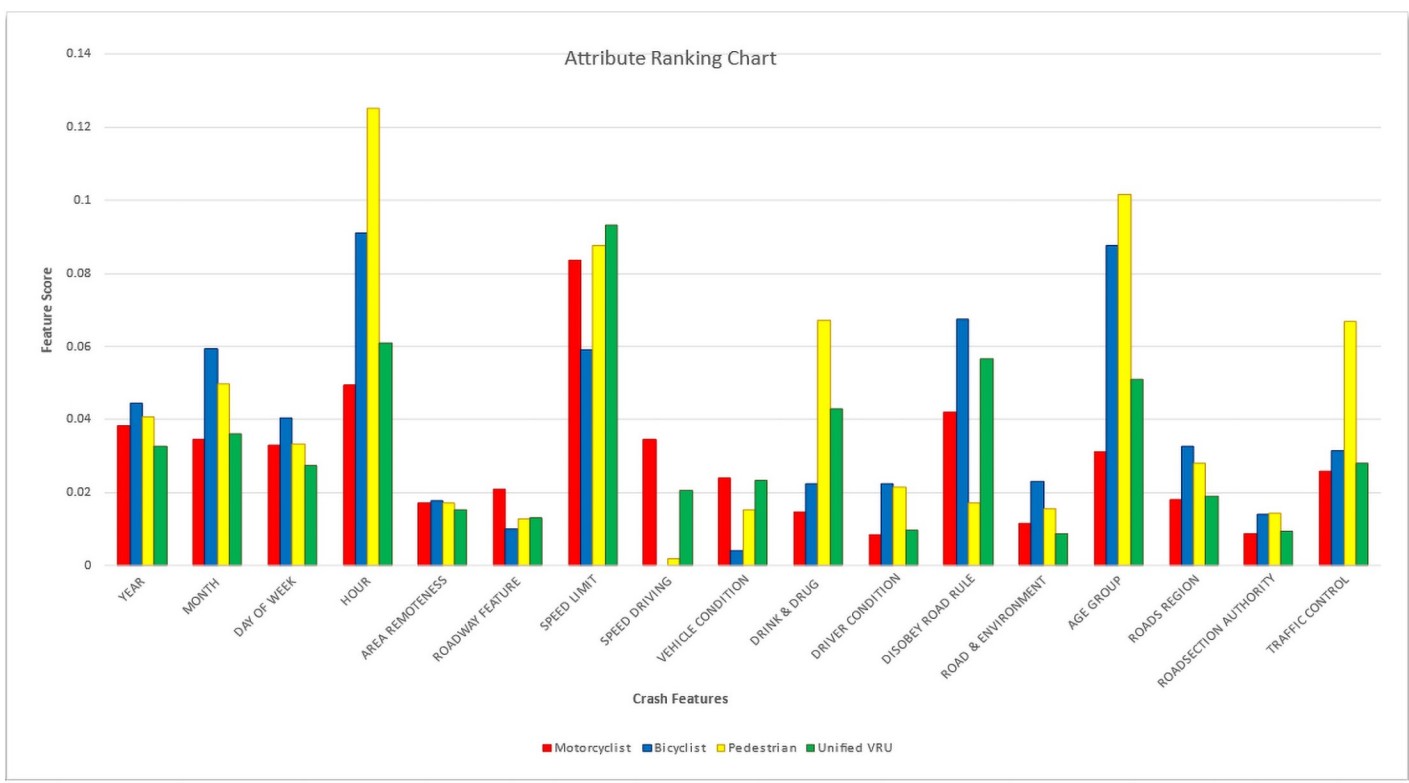

**Fig 4. Random forest based feature ranking for QLD VRU.**

decision tree, the contribution of impurity decreases for each feature in a particular tree is computed. As the first is the combination of trees, the impurity decrease from all the trees are averaged, and the features are ranked accordingly [55, 56]. Feature ranking analysis based on RF is shown in following Fig 4.

In Fig 4, the feature ranking was plotted separately for each of the VRU groups, where the vertical axis presents the feature scores, and the horizontal axis represents the included features for analysis. The features of motorcyclists, bicyclists, pedestrian and unified VRU, are displayed using red, blue, yellow, and green bars, respectively. From the plots, it is observed that hour, posted speed limit and age group are most significant in VRU crash severity analysis as it is drastically higher for all VRU types than any other features. For pedestrians, drink and drug-related crashes and traffic control also impact highly for crash severity classification. Pedestrian and bicyclist of different ages at different times of day show different behaviour on crash severity. Bicyclist crash severity classification has a high impact on road rule violation, and both bicyclist and motorcyclist groups are found highly related to posted speed limit parameter. For the speeding and roadway features, motorcyclist involved crashes were top in the feature ranking, as it shows the highest feature scores compared to the other VRU groups. Moreover, the speed limit factor is found as the top important feature in classifying crash severity of unified VRU. Speeding behaviour also influences motorcyclist crash severity classification, while the speeding factor to any other VRU found to be negligible.

## Crash severity probability analysis

To evaluate the probability for severe crashes in different road crash features, model-based partial dependency plots are generated and discussed in this section. The partial dependence

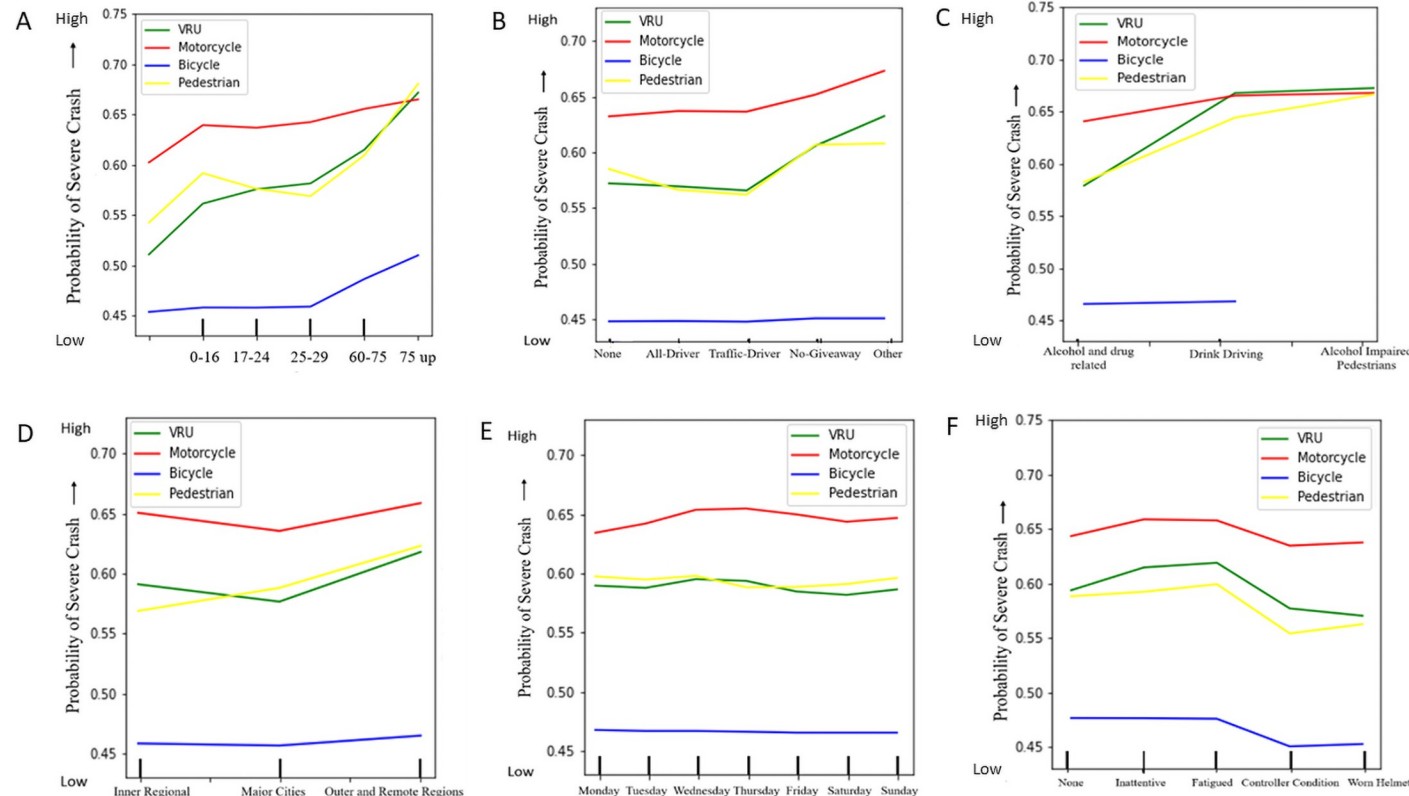

**Fig 5. Partial dependency plots of different features with respect to VRU crash severity.** (A) Age Group (B) Disobey Road Rule (C) Drink, Drug and Alcohol Related (D) Area Remoteness (E) Day of Week (F) Driver Condition.

plots illustrate the marginal dependency of the crash severity on each feature. Here, partial dependency is plotted based on the RF classifier model to demonstrate the impact of individual feature classes over the change of crash severity for all VRU categories of QLD road. Partial plots show the effect of adding one feature (independent variable) to a model, which already contains single or multiple independent features/variables. In this study, the contribution of the different features (independent variables) is interpreted with the corresponding crash severity levels (dependent variables) with the partial plots. Most of the features show a clear indication of their contribution towards the crash severity levels. The probability of crash severity with different road features is interpreted with partial dependency plots as follows in Figs 5–7.

From the partial dependency plots of different road crash features, it is clearly evident that motorcyclist crash severity is extremely higher in almost all road crash feature conditions than any other VRU crash severity. For few road crash features and their subclasses, unified VRU crash severity is found to exceed motorcyclist crash severity. For speeding crashes, unified VRU crash severity slightly surpasses motorcyclist crash severity at the very end. Also, drink driving, alcohol-impaired pedestrians, and unrestrained vehicle condition crashes show a higher severity trend for unified VRU crash severity than motorcyclist severity. For higher posted speed limit crashes (above 80 kmph), unified VRU are found most vulnerable to crash severity and pedestrians with age group near 75, and above are found more likely confronting to high crash severity exceeding both motorcyclist and unified VRU crash severity. Followed by the motorcyclist crash severity, pedestrian severity intervenes with unified VRU crash

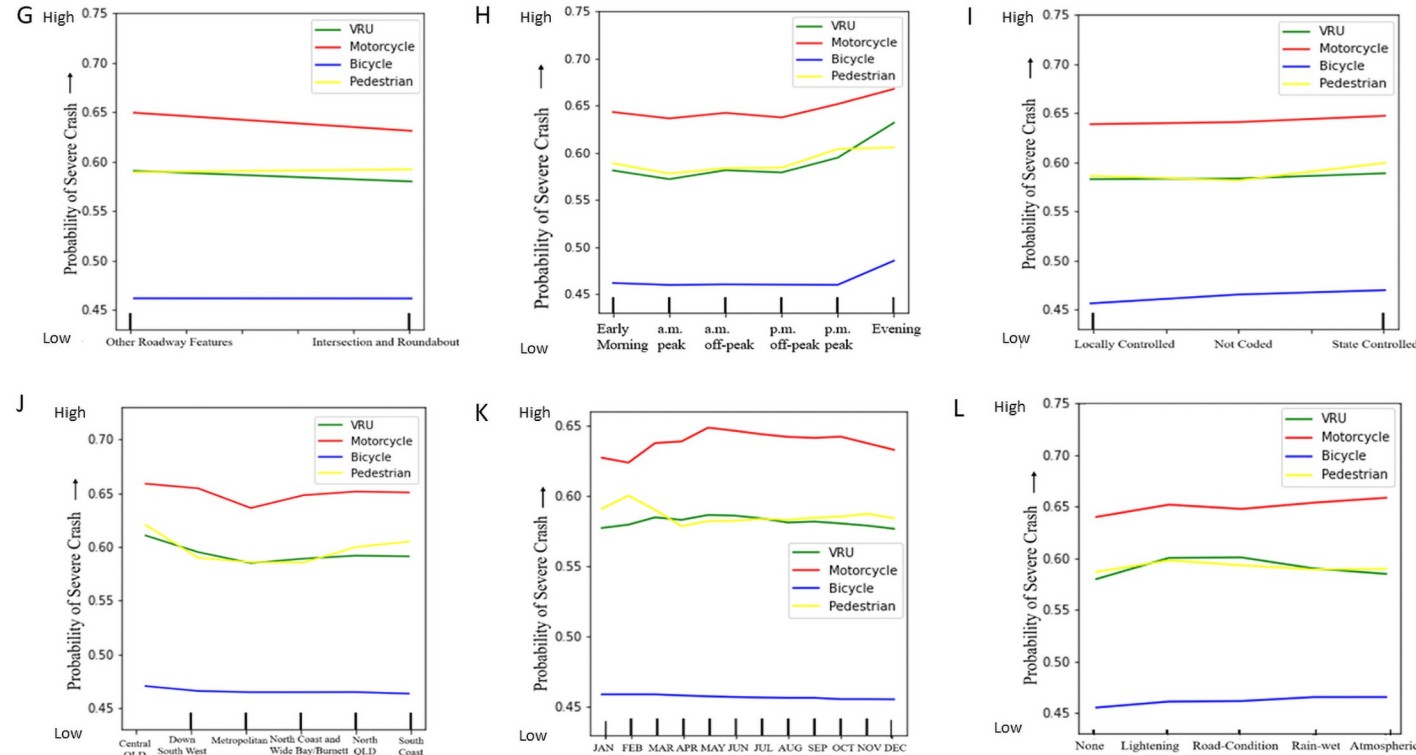

**Fig 6. Partial dependency plots of different features with respect to VRU crash severity.** (G) Roadway Features (H) Hour (Time of Day) (I) Road Section Authority (J) Road Region (K) Month (L) Road and Environment Condition.

severity. Bicyclist crash severity is found comparatively mild in comparison to all other VRU crash severity at Queensland.

For different years and time, the QLD VRU crash severity trend varies significantly. Motorcyclist crash severity is found increasing near 2019, whereas pedestrian crash severity decreases near 2019. Motorcyclist and unified VRU crash severity are found comparatively higher in the middle of weekdays and evening to early morning. All VRU crash severity is comparatively scaled down in major cities and intersections rather than remote areas. Atmospheric condition responses to high motorcyclist crash severity and road condition and lighting condition cause higher unified VRU crash severity. Among different road regions, motorcyclist crash severity is found less in metropolitan areas, and pedestrian crash severity also reduces in Down South West and North Coast regions. Moreover, a drastic drop in pedestrian crash severity is identified under improved traffic control crashes, and bicyclists are found zero crash interaction with alcohol-impaired pedestrians. Unregistered vehicles lead to high crash severity for all VRU. Also, fatigue and inattentive driving conditions are most responsible for all VRU crash severity than any other driving conditions. For all VRU groups, crash severity increases proportionally to the ease of posted speed limit restriction, and this severity trend varies with different road rule violations too.

Overall, motorcyclist crash severity is found comparatively extreme than other VRU types. For all features and their labels, there is a significant difference between motorcyclist crash severity than other VRU types. The pedestrian and unified VRU compete in median position, and still, they are on the verge of higher crash severity levels than the bicyclists, who are found less prone to crash severity than any other VRU types. The partial dependency plots show that

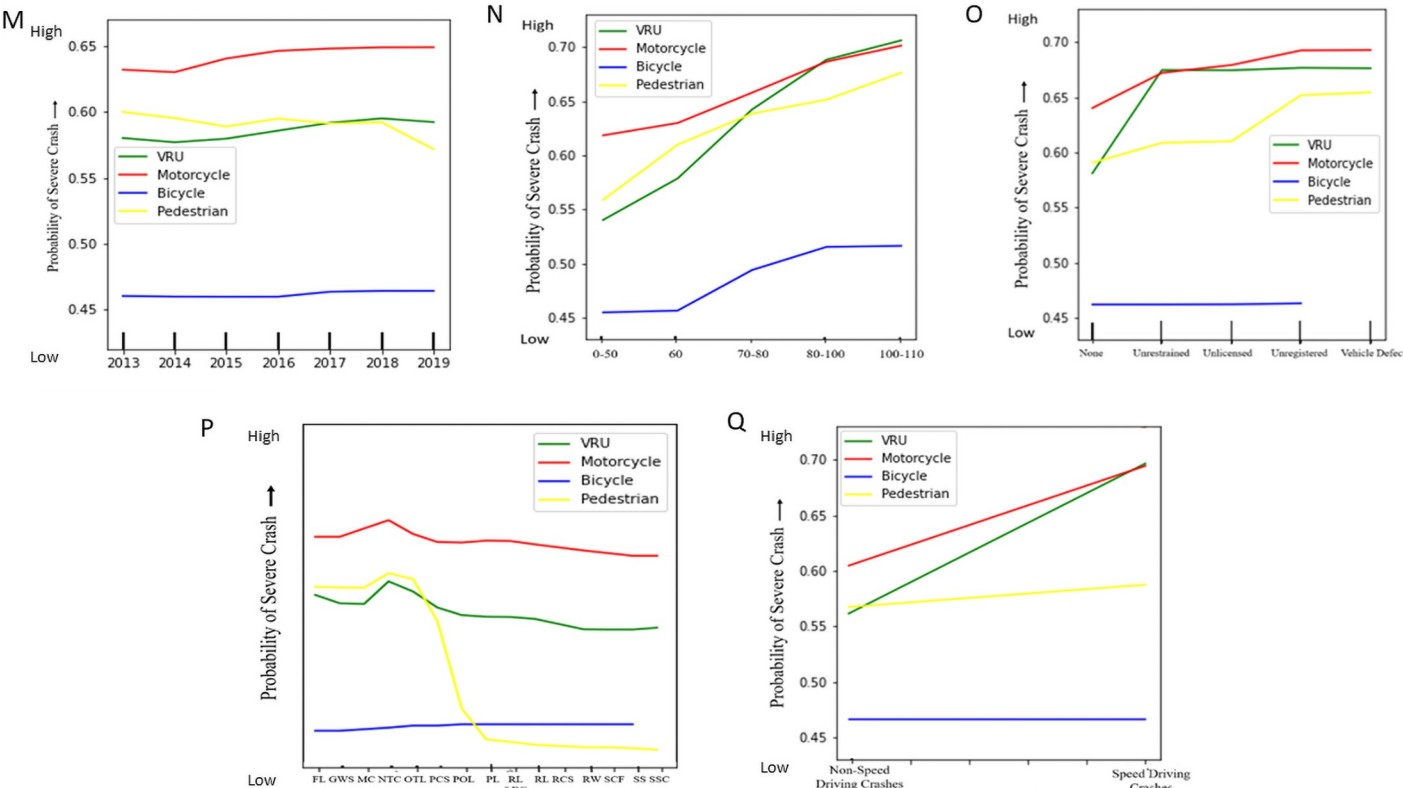

**Fig 7. Partial dependency plots of different features with respect to VRU crash severity.** (M) Year (N) Speed Limit (O) Vehicle Condition (P) Traffic Control (Q) Speed Driving.

VRU crashes are highly affected by several features like age group, speed limits and crash hour. Though there is a major difference found between motorcyclist and bicyclist crash severity level, they show similar trends in variation in most of the plots, whereas the pedestrian-related crash factors are showing a bit different trend, especially in high severity. All VRU types of early age and old age are more vulnerable to the road with higher crash severity, and middle-aged vulnerable road users are found less confronting to severe road crashes.

## Discussion and limitation

This research demonstrates the detailed analysis of VRU crash severity by using crash data from the state of Queensland, Australia, for the years 2013 through to 2019. The factors that highly influence VRU crash severity are identified using machine learning-based classification algorithms RF, SVM and KNN. Also, the most befitting classifier for VRU crash severity classification is evaluated with meticulous feature engineering and consecutive iterations of hyper-parameters. The RF classifier performs the best among the three classifiers, which is consistent with a previous study [36]. Probably this is due to the robustness of the random forest classifier in the large dataset with higher dimensionality. As the RF algorithm works on the ensemble learning based on the voting of multiple decision trees, this model is less prone to overfit, and so the result obtained from the analyses gives a convincing report. Furthermore, the partial dependency plots of each feature provide the depth analysis and relation of each feature with crash severity levels. This also shows a clear distinction of each feature with crash severity levels for different individual VRU groups and unified VRU.

However, some limitations are confronted to achieve higher specificity and accuracy from classification models. With the existing traditional machine learning algorithms, it was a hurdle to get more accuracy keeping all the considerable features. However, the feature ranking and behaviour with respect to VRU crash severity is found quite relevant. Moreover, the sensitivity is found accurate, which refers to the precise classification of high crash severity. Thus, the research fulfils its purpose of analysing VRU crash severity. As deep learning is now getting increasing popularity and giving much utility in the applied machine learning world, future works on this aspect could be done on deep neural network modelling with the existing dataset. For our analysis, we only consider the dataset of road crash data provided by the Department of Transport and Main Road (Queensland). However, the type of collided vehicles (e.g., private cars, vans, trucks or buses) may have a predominant effect on crash severities of vulnerable road users. Such information can be available in the Queensland police records and can be included for the further improvement of crash severity model specification if available. Also, It might be beneficial to compare the performance of discrete choice models with the selected machine learning approaches of our study as a future research avenue.

## Implications

The results found from this study can be used for real-world implications for reducing crash severity of VRU groups. Given that the reported attributes such as crash hour, posted speed limit, age group and traffic control are highly responsible for crashes associated with high severity, the appropriate countermeasures specific to these factors could effectively help to reduce the crash for vulnerable road users. Public awareness and campaign on the given factors could effectively mitigate the risk of VRU crashes. In the peak crash hours, a warning could be given in the specific regions, and the speed limits could be revised during crash peak hours by the law enforcement authority. Speed harmonisation, such as imposing variable speed limit (VSL) using wireless communication based on estimating traffic congestion intensity at peak hours, can help to improve QLD crash severity as well as reduce probable traffic congestions for VRU groups [57]. Also, using police vehicles during historic peak hours with its emergency lights on and maintaining reduced speed triggers other vehicles to maintain harmonic speed causing the reduction of crash severity and traffic congestion [58]. Drink driving and alcohol-impaired pedestrians are still found responsible and vulnerable to higher crash severity. So, more litigation can be imposed to prevent drink driving, and awareness can be raised among people to avoid road crossing in drunk condition. Such preventive measures can be highly ensured in city bar and night club zones where alcohol consumption is regularly higher. Also, inattentive and fatigue driving condition needs frequent observation as they highly trigger road crash severity for VRU. Some advanced technologies and sensors are innovated recently to detect drivers drowsiness [59] which can be effective for drivers. As the elderly peoples are more vulnerable to road crashes, some countermeasures could be taken, such as enforcing an exclusive placard for elderly people aged 75 up (like learner's placard), additional driving training and designing road crossings and footpaths exclusively for the elderly people. Adding more traffic control features could also help to minimise the crash risk for VRU.

## Conclusion

This study contributes towards identifying crash severity factors of different Vulnerable Road User Groups (pedestrian, bicyclist and motorcyclist) while also comparing these factors across different groups. Moreover, the study identified critical factors for all VRU groups together in developing a unified framework to inform road safety solutions. The models we estimated by

employing three different machine learning algorithms—RF, SVM and KNN by using data from Queensland, Australia, for the years 2013 through to 2019. The identification of the impact of different features on VRU crash severity with respect to crash severity levels is practically crucial in future planning and improvement of QLD road for vulnerable road users. Among three machine learning algorithms, the random forest-based classification model was found to perform better relative to other algorithms while getting insight into the contribution of the features on crash severity for different VRU. Moreover, this research analysed the contribution of each feature on the crash severity levels with partial plots and feature importance wrapping with random forest. Thus, the latest condition of QLD road for VRU was compared to identify the most critical condition of QLD VRU crash severity features. By scrutinising the most critical condition of these features and their time, the probable VRU friendly factors can be distinguished and used to improve QLD roads for VRU. The result analysis shows higher motorcyclist crash severity among VRU groups for any road crash parameter conditions at QLD. The pedestrians and unified VRU are also highly vulnerable to severe crashes. Only bicyclist crash severity is found comparatively mild than other VRU at QLD.

## Acknowledgments

The authors would like to acknowledge the Department of Transport and Main Road (TMR), Australia, for providing the road crash data for this study.

## Author Contributions

**Conceptualization:** Md Mostafizur Rahman Komol, Mohammed Elhenawy, Shamsunnahar Yasmin.

**Data curation:** Md Mostafizur Rahman Komol, Mohammed Elhenawy.

**Formal analysis:** Md Mostafizur Rahman Komol, Mohammed Elhenawy, Shamsunnahar Yasmin.

**Funding acquisition:** Md Mostafizur Rahman Komol, Mohammed Elhenawy, Andry Rakotonirainy.

**Investigation:** Md Mostafizur Rahman Komol, Mohammed Elhenawy, Shamsunnahar Yasmin.

**Methodology:** Md Mostafizur Rahman Komol, Md Mahmudul Hasan, Mohammed Elhenawy, Shamsunnahar Yasmin.

**Project administration:** Md Mostafizur Rahman Komol.

**Resources:** Md Mostafizur Rahman Komol.

**Software:** Md Mostafizur Rahman Komol, Mohammed Elhenawy.

**Supervision:** Mohammed Elhenawy, Shamsunnahar Yasmin, Mahmoud Masoud, Andry Rakotonirainy.

**Validation:** Md Mostafizur Rahman Komol.

**Visualization:** Md Mostafizur Rahman Komol, Shamsunnahar Yasmin.

**Writing – original draft:** Md Mostafizur Rahman Komol, Md Mahmudul Hasan, Shamsunnahar Yasmin.

**Writing – review & editing:** Md Mostafizur Rahman Komol, Md Mahmudul Hasan, Mohammed Elhenawy, Shamsunnahar Yasmin.

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
