## [Decision Letter · Decision Letter 0]

10 Jun 2021

PONE-D-21-14520

Crash Severity Analysis of Vulnerable Road Users using Machine Learning

PLOS ONE

Dear Dr. Komol,

Thank you for submitting your manuscript to PLOS ONE. After careful consideration, we feel that it has merit but does not fully meet PLOS ONE’s publication criteria as it currently stands. Therefore, we invite you to submit a revised version of the manuscript that addresses the points raised during the review process.

We look forward to receiving your revised manuscript.

Kind regards,

Feng Chen

Academic Editor

PLOS ONE

Journal Requirements:

2)  We note that you have indicated that data from this study are available upon request. PLOS only allows data to be available upon request if there are legal or ethical restrictions on sharing data publicly. For more information on unacceptable data access restrictions, please see http://journals.plos.org/plosone/s/data-availability#loc-unacceptable-data-access-restrictions.

3) We noted in your submission details that a portion of your manuscript may have been presented or published elsewhere. [Yes. we have submitted the abstract of this paper for publication at a conference call for abstract only: ASCE International Conference on Transportation and Development. The abstract was accepted at the conference. It was considered an abstract only submission, not a full paper publication. Also, they will not take the copyright. So, it is safe for consideration under PLOS ONE publication].

Please clarify whether this conference proceeding or publication was peer-reviewed and formally published. If this work was previously peer-reviewed and published, in the cover letter please provide the reason that this work does not constitute dual publication and should be included in the current manuscript.

Reviewers' comments:

Reviewer's Responses to Questions

**Comments to the Author**

1. Is the manuscript technically sound, and do the data support the conclusions?

Reviewer #1: Partly

Reviewer #2: Yes

2. Has the statistical analysis been performed appropriately and rigorously? 

Reviewer #1: Yes

Reviewer #2: Yes

3. Have the authors made all data underlying the findings in their manuscript fully available?

Reviewer #1: No

Reviewer #2: Yes

4. Is the manuscript presented in an intelligible fashion and written in standard English?

Reviewer #1: Yes

Reviewer #2: Yes

5. Review Comments to the Author

Reviewer #1: This study investigated the injury severity of vulnerable road users involved in traffic crashes using three supervised machine learning algorithms namely K-nearest neighbor (KNN), support vector machine (SVM), and random forest (RF). Overall, the topic is interesting and worthy of investigation. The whole manuscript is also well organized and easy to follow. Before suggesting it for publication, several issues, however, need to be well addressed.

1. Among the various machine learning methods, why were the three algorithms namely KNN, SVM, and RF used for analysis? More justifications on the advancement of these three methods are required in the Introduction and Literature Review sections.

2. The discrete choice models have long been used for crash severity analysis (Mannering et al., 2016). More importantly, the random-parameter models have been demonstrated to be superior in accounting for the unobserved heterogeneity, with a substantial improvement in goodness-of-fit and interpretability (Chang et al., 2019; Chen et al., 2019; Waseem et al., 2019; Alogaili and Mannering, 2020; Wang et al., 2020; Zhou et al., 2020; Zhou et al., 2021). These methodological alternatives should not be ignored, particularly in the Literature Review section. In addition, since the random-parameter logit model has become the benchmark for crash data analysis (Mannering and Bhat, 2014), the authors are highly suggested to include this method for comparison, which may help to further highlight the advantages of machine learning methods.

3. When investigating the safety of vulnerable road users, the type of collided vehicles (e.g., private cars, vans, trucks or buses) is expected to have a predominant effect on crash severities. Such information is readily available in the police records and should thus be included for model specification.

4. There are typos throughout the manuscript. For example, on page 9 line 194, “this help understanding…” should be “this helps in understanding…”; on page 10 line 221, “bicyclist” and “pedestrian” should be “bicyclists” and “pedestrians”; on page 24 line 286, “four different classifiers” should be “three different classifiers”.

References

Alogaili and Mannering, 2020. Unobserved heterogeneity and the effects of driver nationality on crash injury severities in Saudi Arabia. Accident Analysis & Prevention 144, 105618.

Chang et al., 2019. Investigating injury severities of motorcycle riders: a two-step method integrating latent class cluster analysis and random parameters logit model. Accident Analysis & Prevention 131, 316-326.

Chen et al., 2019. Investigation on the injury severity of drivers in rear-end collisions between cars using a random parameters bivariate ordered probit model. International Journal of Environmental Research and Public Health 16(14), 2632.

Mannering and Bhat, 2014. Analytic methods in accident research: methodological frontier and future directions. Analytic Methods in Accident Research 1, 1-22.

Mannering et al., 2016. Unobserved heterogeneity and the statistical analysis of highway accident data. Analytic Methods in Accident Research 11, 1-16.

Wang et al., 2020. Random parameter probit models to analyze pedestrian red-light violations and injury severity in pedestrian–motor vehicle crashes at signalized crossings. Journal of Transportation Safety & Security 12(6), 818-837.

Waseem et al., 2019. Factors affecting motorcyclists’ injury severities: an empirical assessment using random parameters logit model with heterogeneity in means and variances. Accident Analysis & Prevention 123, 12-19.

Zhou et al., 2020. Severity of passenger injuries on public buses: a comparative analysis of collision injuries and non-collision injuries. Journal of Safety Research 74, 55-69.

Zhou et al., 2021. Factors associated with consecutive and non-consecutive crashes on freeways: a two-level logistic modeling approach. Accident Analysis & Prevention 154, 106054.

Reviewer #2: The current study attempts to examine the injury severity levels of VRUs using machine learning models. The topic is interesting and the paper is overall well-written. There are some issues to be addressed before it can be accepted for publication:

1. for injury severity studies, the choice between statistical models and machine learning ones is often a tradeoff between model interpretation and predictive performance. The authors should add discussion about the interpretability of the chosen machine learning models.

2. The literature review should be strengthened, the following relevant severity studies should be acknowledged and discussed in the paper:

[1] Shao Xiaojun, Ma Xiaoxiang, Chen Feng, Song Mingtao, Pan Xiaodong, You Kesi, 2020. A random parameters ordered probit analysis of injury severity in truck involved rear-end collisions. International journal of environmental research and public health. doi:10.3390/ijerph17020395

[2] Dong Bowen, Ma Xiaoxiang, Chen Feng, 2018. Analyzing the Injury Severity Sustained by Non-motorists at Mid-block Considering Non-motorists’ Pre-crash Behavior, Transportation Research Record.

6. PLOS authors have the option to publish the peer review history of their article (what does this mean?). If published, this will include your full peer review and any attached files.

Reviewer #1: No

Reviewer #2: No

---

## [Author Response · Author response to Decision Letter 0]

9 Jul 2021

Paper Title: Crash Severity Analysis of Vulnerable Road Users using Machine Learning

Response to the Reviewers’ Comments

Manuscript ID: PONE-D-21-14520

We would like to thank the reviewers for their valuable comments. We believe that the comments have identified important areas that required improvement. We have addressed the comments point by point and revised the paper accordingly. Please note that our responses to the comments are highlighted in the context of the paper in a different colour or track changes.

Reviewers’ comments:

#1 Submitted by: Reviewer 1

Comments:

1. Among the various machine learning methods, why were the three algorithms, namely KNN, SVM, and RF, used for analysis? More justifications on the advancement of these three methods are required in the Introduction and Literature Review sections.

Response:

The justification for using these models is added in the introduction section of the revised paper.

So far, there is no specified rule to apply a particular machine learning model for a specific application [9]. Therefore, most of the research uses multiple machine learning models to the same dataset and use comparative analysis. For example, the study performed by Thanh Noi and Kappas [10] compared the performance of three supervised machine learning models, i.e., KNN, RFC and SVM, in the classification of remote sensing images. The authors found the SVM classifier to be the best performing model in this application, while considering accuracy as performance measure. This indicate that different machine learning models show promises in different applications; however, which classifier is best for a specific application is still not clear. 

Specifically, in the existing road safety and crash severity analysis research, several studies have compared the performance between machine learning models where KNN, SVM, and RF models show their performance advantages at different domains [11-13]. Therefore, all of these models bear the significance to be used in machine learning-based analysis for road crash severity. These models perform better in different applications, and it is not feasible to conclude superiority of one without comparing their performance for the specific application.

Reference:

9. Chen J, Wang H, Hua C. Assessment of driver drowsiness using electroencephalogram signals based on multiple functional brain networks. International Journal of Psychophysiology. 2018;133:120-30.

10. Thanh Noi P, Kappas M. Comparison of random forest, k-nearest neighbor, and support vector machine classifiers for land cover classification using Sentinel-2 imagery. Sensors. 2018;18(1):18.

11. Elamrani Abou Elassad Z, Mousannif H, Al Moatassime H. A real-time crash prediction fusion framework: An imbalance-aware strategy for collision avoidance systems. Transportation Research Part C: Emerging Technologies. 2020;118:102708.

12. Zhang J, Li Z, Pu Z, Xu C. Comparing Prediction Performance for Crash Injury Severity Among Various Machine Learning and Statistical Methods. IEEE Access. 2018;6:60079-87.

13. Fiorentini N, Losa M. Handling Imbalanced Data in Road Crash Severity Prediction by Machine Learning Algorithms. Infrastructures. 2020;5(7).

2. The discrete choice models have long been used for crash severity analysis (Mannering et al., 2016). More importantly, the random-parameter models have been demonstrated to be superior in accounting for the unobserved heterogeneity, with a substantial improvement in goodness-of-fit and interpretability (Chang et al., 2019; Chen et al., 2019; Waseem et al., 2019; Alogaili and Mannering, 2020; Wang et al., 2020; Zhou et al., 2020; Zhou et al., 2021). These methodological alternatives should not be ignored, particularly in the Literature Review section. In addition, since the random-parameter logit model has become the benchmark for crash data analysis (Mannering and Bhat, 2014), the authors are highly suggested to include this method for comparison, which may help to further highlight the advantages of machine learning methods.

Response: We have added the following context and references in the literature review section to address the reviewer’s concern. 

The suggested comparison between the discrete choice model and machine learning is beyond the scope of this study since our main focus was on comparing and contrasting effects of factors on different vulnerable road user groups safety by employing a machine learning approach. However, we appreciate the kind suggestions, and we have added the suggested comparison as a future research avenue in the conclusion section of the revised manuscript.

Discrete choice model, such as random parameter model, random parameter ordered probit analysis, random parameter logit model, mixed logic model are most common statistical approaches used in different road crash data analysis [23-33].

References:

23. Dong B, Ma X, Chen F. Analyzing the Injury Severity Sustained by Non-Motorists at Mid-Blocks considering Non-Motorists’ Pre-Crash Behavior. Transportation Research Record. 2018;2672(38):138-48.

24. Shao X, Ma X, Chen F, Song M, Pan X, You K. A Random Parameters Ordered Probit Analysis of Injury Severity in Truck Involved Rear-End Collisions. International Journal of Environmental Research and Public Health. 2020;17(2).

25. Alogaili A, Mannering F. Unobserved heterogeneity and the effects of driver nationality on crash injury severities in Saudi Arabia. Accident Analysis & Prevention. 2020;144:105618.

26. Chang F, Xu P, Zhou H, Chan AHS, Huang H. Investigating injury severities of motorcycle riders: A two-step method integrating latent class cluster analysis and random parameters logit model. Accident Analysis & Prevention. 2019;131:316-26.

27. Chen F, Song M, Ma X. Investigation on the Injury Severity of Drivers in Rear-End Collisions Between Cars Using a Random Parameters Bivariate Ordered Probit Model. International Journal of Environmental Research and Public Health. 2019;16(14).

28. Mannering FL, Bhat CR. Analytic methods in accident research: Methodological frontier and future directions. Analytic Methods in Accident Research. 2014;1:1-22.

29. Mannering FL, Shankar V, Bhat CR. Unobserved heterogeneity and the statistical analysis of highway accident data. Analytic Methods in Accident Research. 2016;11:1-16.

30. Wang J, Huang H, Xu P, Xie S, Wong SC. Random parameter probit models to analyze pedestrian red-light violations and injury severity in pedestrian–motor vehicle crashes at signalized crossings. Journal of Transportation Safety & Security. 2020;12(6):818-37.

31. Waseem M, Ahmed A, Saeed TU. Factors affecting motorcyclists’ injury severities: An empirical assessment using random parameters logit model with heterogeneity in means and variances. Accident Analysis & Prevention. 2019;123:12-9.

32. Zhou H, Yuan C, Dong N, Wong SC, Xu P. Severity of passenger injuries on public buses: A comparative analysis of collision injuries and non-collision injuries. Journal of Safety Research. 2020;74:55-69.

33. Zichu Z, Fanyu M, Cancan S, Richard T, Zhongyin G, Lili Y, et al. Factors associated with consecutive and non-consecutive crashes on freeways: A two-level logistic modeling approach. Accident Analysis & Prevention. 2021;154:106054.

In the Discussion and Limitation section-

“It might be beneficial to compare the performance of discrete choice models with the selected machine learning approaches of our study as a future research avenue.” 

3. When investigating the safety of vulnerable road users, the type of collided vehicles (e.g., private cars, vans, trucks or buses) is expected to have a predominant effect on crash severities. Such information is readily available in the police records and should thus be included for model specification.

Response: For our study, we focused only on Queensland road crash data provided by the Department of Transport and Main Road (Queensland). We have added your suggested idea in the Discussion and Limitation Section of the updated manuscript.

“For our analysis, we only consider the dataset of road crash data provided by Department of Transport and Main Road (Queensland). However, the type of collided vehicles (e.g., private cars, vans, trucks or buses) may have a predominant effect on crash severities of vulnerable road users. Such information can be available in the Queensland police records and can be included for the further improvement of crash severity model specification if available.”

4. There are typos throughout the manuscript. For example, on page 9 line 194, “this help understanding…” should be “this helps in understanding…”; on page 10 line 221, “bicyclist” and “pedestrian” should be “bicyclists” and “pedestrians”; on page 24 line 286, “four different classifiers” should be “three different classifiers”.

Response: Thanks a lot for indicating the corrections. We proofread the whole document and revised the manuscript carefully in addressing the concerns. Also, we have corrected Figure 2, which had some typos before. 

#2 Submitted by: Reviewer 2

Comments:

1. for injury severity studies, the choice between statistical models and machine learning ones is often a tradeoff between model interpretation and predictive performance. The authors should add discussion about the interpretability of the chosen machine learning models.

Response: We appreciate your kind comment. We have added the following context to discuss model architecture in section Classification Approaches.

Most of the statistical models work to infer the relationship between or among two or more variables, whereas the machine learning models deal with making the best accurate predictions [50]. The predictor outcomes are used to identify the class of the target variables. On the one hand, the statistical models can handle very small datasets, whereas the machine learning models are designed to deal with big data [51]. The dataset contains 21158 data points, having 17 features. Using statistical methods, such as ANOVA F Test or Correlation-based methods, could make the interpretation complex; thus, we have used machine learning models in this study. Given the different machine learning-based predictive models works on their specific principles, it is worth comparing their performances on the same dataset. The three selected models are supervised learning algorithms. SVM is a linear model that identifies the best classification hyperplane to separate data into desired classes [52]. KNN algorithm analyses the similarity of the data to classify different classes [53]. It considers a K value at the nearest neighbour to the data point for classification. The random forest acts as an ensemble method bagging multiple decision trees for classification [54]. The decision tree is an algorithm containing series of trees with the binary decision about the class. As such, random forest outperforms decision tree as an ensemble technique, decision tree model is not separately used for our crash severity analysis.

References:

50. Tang J, Zheng L, Han C, Yin W, Zhang Y, Zou Y, et al. Statistical and machine-learning methods for clearance time prediction of road incidents: A methodology review. Analytic Methods in Accident Research. 2020;27:100123.

51. Makridakis S, Spiliotis E, Assimakopoulos V. Statistical and Machine Learning forecasting methods: Concerns and ways forward. PLOS ONE. 2018;13(3):e0194889.

52. Cristianini N, Ricci E. Support Vector Machines. In: Kao M-Y, editor. Encyclopedia of Algorithms. New York, NY: Springer New York; 2016. p. 2170-4.

53. Peterson LE. K-Nearest Neighbour. Scholarpedia. 2009;4(2).

54. Pal M. Random forest classifier for remote sensing classification. International Journal of Remote Sensing. 2005;26(1):217-22.

2. The literature review should be strengthened, the following relevant severity studies should be acknowledged and discussed in the paper:

[1] Shao Xiaojun, Ma Xiaoxiang, Chen Feng, Song Mingtao, Pan Xiaodong, You Kesi, 2020. A random parameters ordered probit analysis of injury severity in truck involved rear-end collisions. International journal of environmental research and public health. doi:10.3390/ijerph17020395

[2] Dong Bowen, Ma Xiaoxiang, Chen Feng, 2018. Analyzing the Injury Severity Sustained by Non-motorists at Mid-block Considering Non-motorists’ Pre-crash Behavior, Transportation Research Record.

Response: Thanks for the comment. We have added these references in literature review section and the following context was added.

Discrete choice model, such as random parameter model, random parameter ordered probit analysis, random parameter logit model, mixed logic model are most common statistical approaches used in different road crash data analysis [23-33].

References:

23. Dong B, Ma X, Chen F. Analyzing the Injury Severity Sustained by Non-Motorists at Mid-Blocks considering Non-Motorists’ Pre-Crash Behavior. Transportation Research Record. 2018;2672(38):138-48.

24. Shao X, Ma X, Chen F, Song M, Pan X, You K. A Random Parameters Ordered Probit Analysis of Injury Severity in Truck Involved Rear-End Collisions. International Journal of Environmental Research and Public Health. 2020;17(2).

---

## [Decision Letter · Decision Letter 1]

26 Jul 2021

Crash Severity Analysis of Vulnerable Road Users using Machine Learning

PONE-D-21-14520R1

Dear Dr. Komol,

We’re pleased to inform you that your manuscript has been judged scientifically suitable for publication and will be formally accepted for publication once it meets all outstanding technical requirements.

Kind regards,

Feng Chen

Academic Editor

PLOS ONE

Additional Editor Comments (optional):

Reviewers' comments:

Reviewer's Responses to Questions

**Comments to the Author**

1. If the authors have adequately addressed your comments raised in a previous round of review and you feel that this manuscript is now acceptable for publication, you may indicate that here to bypass the “Comments to the Author” section, enter your conflict of interest statement in the “Confidential to Editor” section, and submit your "Accept" recommendation.

Reviewer #1: All comments have been addressed

2. Is the manuscript technically sound, and do the data support the conclusions?

Reviewer #1: Partly

3. Has the statistical analysis been performed appropriately and rigorously? 

Reviewer #1: Yes

4. Have the authors made all data underlying the findings in their manuscript fully available?

Reviewer #1: No

5. Is the manuscript presented in an intelligible fashion and written in standard English?

Reviewer #1: Yes

6. Review Comments to the Author

Reviewer #1: I have no further comments, as the authors have adequately addressed my concerns. I therefore suggest it for publication.

7. PLOS authors have the option to publish the peer review history of their article (what does this mean?). If published, this will include your full peer review and any attached files.

Reviewer #1: No

---

## [Editor Report · Acceptance letter]

29 Jul 2021

PONE-D-21-14520R1 

Crash Severity Analysis of Vulnerable Road Users using Machine Learning 

Dear Dr. Komol:

I'm pleased to inform you that your manuscript has been deemed suitable for publication in PLOS ONE. Congratulations! Your manuscript is now with our production department. 

Kind regards, 

on behalf of

Dr. Feng Chen 

Academic Editor

PLOS ONE